# Infected *Ixodes scapularis* Nymphs Maintained in Prolonged Questing under Optimal Environmental Conditions for One Year Can Transmit *Borrelia burgdorferi* (*Borreliella genus novum*) to Uninfected Hosts

Kamalika Samanta,[a] Jose F. Azevedo,[b] Nisha Nair,[b] Suman Kundu,[b] Maria Gomes-Solecki[a,b]

[a]Department of Pharmaceutical Sciences, University of Tennessee Health Science Center, Memphis, Tennessee, USA
[b]Department of Microbiology, Immunology, and Biochemistry, University of Tennessee Health Science Center, Memphis, Tennessee, USA

**ABSTRACT** In recent decades, Lyme disease has been expanding to previous nonendemic areas. We hypothesized that infected *I. scapularis* nymphs that retain host-seeking behavior under optimal environmental conditions are fit to fulfil their transmission role in the enzootic cycle of *B. burgdorferi*. We produced nymphal ticks in the laboratory under controlled temperature (22–25°C), humidity (80–90%), and natural daylight cycle conditions to allow them to retain host-seeking/questing behavior for 1 year. We then analyzed differences in *B. burgdorferi* infection prevalence in questing and diapause nymphs at 6 weeks postmolting (prime questing) as well as differences in infection prevalence of questing nymphs maintained under prolonged environmental induced questing over 12 months (prolonged questing). Lastly, we analyzed the fitness of nymphal ticks subjected to prolonged questing in transmission of *B. burgdorferi* to naive mice over the course of the year. *B. burgdorferi* infected unfed *I. scapularis* nymphal ticks maintained under optimal environmental conditions in the laboratory not only survived for a year in a developmental state of prolonged questing (host-seeking), but they retained an infection prevalence sufficient to effectively fulfil transmission of *B. burgdorferi* to uninfected mice after tick challenge. Our study is important for understanding and modeling Lyme disease expansion into former nonendemic regions due to climate change.

**IMPORTANCE** Lyme disease is rapidly spreading from its usual endemic areas in the Northeast, Midwest, and Midatlantic states into neighboring areas, which could be due to changing climate patterns. Our study shows that unfed *I. scapularis* nymphal ticks kept under optimal environmental conditions in the laboratory survived for a year while exhibiting aggressive host-seeking behavior, and they maintained a *B. burgdorferi* infection prevalence which was sufficient to infect naive reservoir hosts after tick challenge. Our study raises important questions regarding prolonged survival of *B. burgdorferi* infected host-seeking nymphal *I. scapularis* ticks that can potentially increase the risk of Lyme disease incidence, if conditions of temperature and humidity become amenable to the enzootic cycle of *B. burgdorferi* in regions currently classified as nonendemic.

**KEYWORDS** *Borrelia burgdorferi*, *Borreliella*, diapause, host-seeking, Ixodes scapularis, Lyme disease, questing, tick challenge

*I*xodes scapularis*, commonly known as blacklegged deer tick, is a major transmission vector of pathogens that cause diseases such as anaplasmosis, babesiosis, *Borrelia miyamotoi* disease, Powassan viral disease, ehrlichiosis, and Lyme disease (1). *B. burgdorferi sensu lato*, which causes Lyme disease (or Lyme borreliosis), transits between vertebrate reservoir hosts and *I. scapularis* tick vectors in an endless enzootic cycle. In recent decades, Lyme disease has been expanding to previously nonendemic areas (2, 3). One reason for this expansion may be

Address correspondence to Maria Gomes-Solecki, mgomesso@uthsc.edu.

The authors declare a conflict of interest. The following authors declare potential conflicts of interest: M.G.-S. (grants from federal agencies, employment). All other authors declare no conflicts.

changes in environmental conditions better suited to the maintenance of the enzootic cycle of *B. burgdorferi* (4–6).

The seasonal pattern of host-seeking behavior (questing) of *I. scapularis* determines the relative risk of tick bites (7) and the consequent transmission of *B. burgdorferi* and Lyme disease. This seasonal activity varies slightly between the northeastern, northcentral, and southern regions of the United States. In the northeast, adult ticks are mostly active in the fall, nymphal tick activity increases in spring and early summer, and larval tick activity peaks in mid-summer (8); in northcentral and southern regions, adult ticks are active in the fall/winter, and nymphal and larval tick activity overlaps in the spring/summer (9–11). There is general agreement that ticks avoid host-seeking behavior (questing) at unfavorable times of the year, when the temperatures are too high or too low and the humidity is too low (12). Thus, ticks maximize survival by responding to environmental conditions. It has been shown that *I. scapularis* nymphs remain viable for 10 months in optimal environmental laboratory conditions of temperature, humidity, and natural light (22–24°C, 90%, 16 h/8 h light/dark), but whether they can effectively transmit *B. burgdorferi* to the mammalian host was not tested (13). In addition, ticks lengthen their life cycle by entering various forms of dormancy such as behavioral (quiescent) and developmental diapause (14). Survival of *B. burgdorferi* in unfed ticks throughout long periods of diapause is important for continuity of the enzootic cycle. However, in *I. scapularis*, it is not known whether the numbers of *B. burgdorferi* are affected as the ticks undergo diapause and questing.

Stadial *I. scapularis* nymphs are critical to the maintenance of *B. burgdorferi* in the enzootic cycle because they transmit *B. burgdorferi* to uninfected hosts, such as reservoir and incidental hosts (15, 16). Some studies found that countries with high incidence of Lyme borreliosis (Austria, Czech Republic, Slovenia, Switzerland, and Sweden) have high *B. burgdorferi sensu lato* prevalence in nymphal ticks, whereas countries with low incidence of Lyme disease (France, Great Britain, Italy, and Poland) have a low prevalence of *B. burgdorferi* in collected ticks (16–18).

We hypothesized that *I. scapularis* nymphs kept in prolonged questing by optimal environmental conditions are fit to fulfil their transmission role in the enzootic cycle of *B. burgdorferi*. We produced nymphal ticks in the laboratory under controlled temperature, humidity, and natural light/dark cycle conditions to allow them to remain in host-seeking (questing) phase for 1 year. We then analyzed differences in *B. burgdorferi* infection prevalence of questing and diapause nymphs at 6 weeks postmolting (prime questing) as well as differences in infection prevalence of questing nymphs maintained under prolonged environmentally induced questing over 12 months (prolonged questing). Lastly, we analyzed the fitness of nymphal ticks subjected to environmental induced questing in transmission of *B. burgdorferi* to naive mice over the course of the year.

## RESULTS

***B. burgdorferi* infection prevalence was significantly higher in questing than in diapause nymphs.** Upon observation of our nymphs at the beginning of prime feeding time, 6 weeks postmolting from larva, we noticed considerable differences regarding their mobility behavior. Thus, before quantification of *B. burgdorferi* infection by qPCR, we sorted nymphal ticks from the same glass vial into two groups: (i) in diapause, low or no mobility, congregated at the bottom of the vial; and (ii) host-seeking (questing), highly mobile, gathered on the mesh at the top of the vial. *B. burgdorferi* flaB qPCR showed that 3 out of 12 ticks (25%) exhibiting diapause behavior were positive with an average *B. burgdorferi* load of $4.41 \times 10^2$ and that 9 out of 12 host-seeking nymphs (75%) were positive with an average *B. burgdorferi* load of $3.2 \times 10^3$ (Fig. 1). Differences in load were statistically significant ($P = 0.0065$).

**Nymphal infection prevalence (NIP) of ticks kept in host-seeking behavior by optimal environmental conditions remained high for up to 1 year.** *I. scapularis* nymphs kept in prolonged questing by optimal conditions of temperature (22–24°C), humidity (80–90%), and natural daylight were checked monthly ($n = 10$–12 per month) by *B. burgdorferi* flaB qPCR (Fig. 2) from June 2020 to June 2021. Nymphal infection prevalence started at 75% in June 2020, 6 weeks after molting, it fluctuated randomly

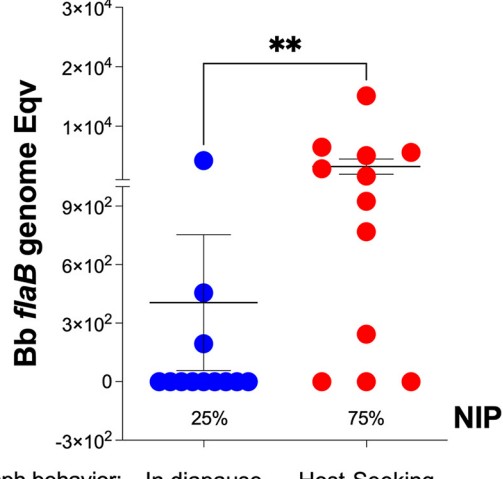

**FIG 1** *B. burgdorferi* load and nymphal infection prevalence (NIP) of flat *I. scapularis* 6 weeks postmolting. *n* = 12 per group. Statistics by Mann-Whitney *t* test. **, *P* = 0.0065.

between 40% to 100% throughout the year, and at conclusion of the experiment it was 60%, in June 2021, ~15 months after molting from infected larva. Thus, laboratory-maintained questing did not negatively affect NIP for ~1 year. Differences in *B. burgdorferi* load compared month to month were significantly increased between June and July 2020 (*P* = 0.0242), and they were decreased in March–April 2021 (*P* = 0.0050).

**Nymphal infection prevalence of the engorged *I. scapularis* used for tick challenges declined in the last quarter of the year.** Five independent tick challenges were performed throughout the year using the nymphal ticks maintained in questing under optimal environmental conditions in the laboratory. After tick challenge, fully engorged *I. scapularis* ticks were retrieved from the bottom of the cages after naturally falling off. PCR quantification of *B. burgdorferi* load from each tick (*n* = 12 to 19 per group) ranged between $5 \times 10^3$ and $4 \times 10^5$ flaB genomes (Fig. 3); differences were not statistically significant. Regarding nymphal

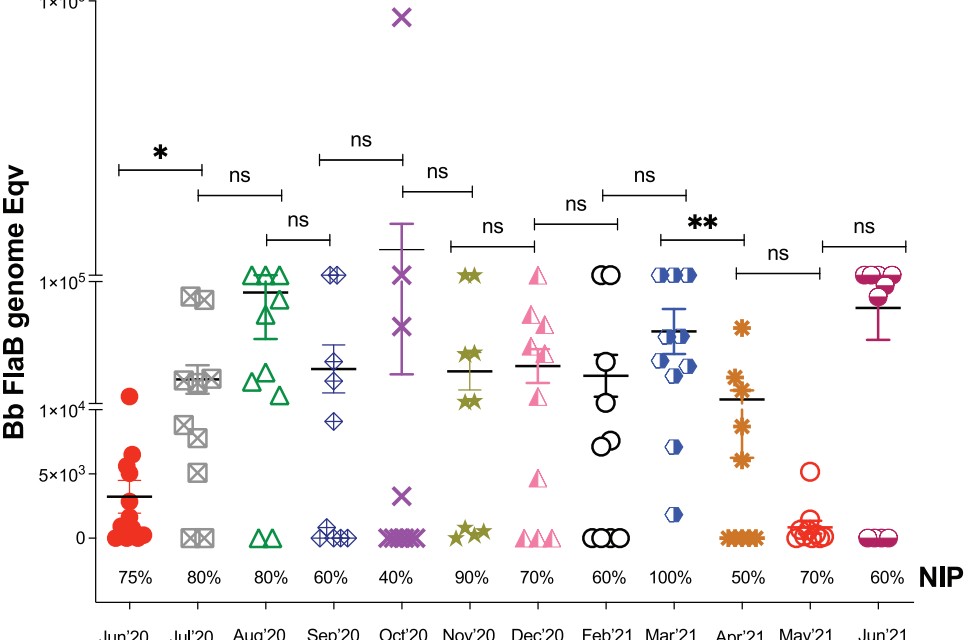

**FIG 2** *B. burgdorferi* load and nymphal infection prevalence (NIP) of host-seeking (questing) flat *I. scapularis* ticks tested each month for 13 months. Data represent mean ± SEM. *n* = 10–12 per group. Statistics by Mann-Whitney *t* test. **, *P* = 0.0050; *, *P* = 0.0242; ns = not significant.

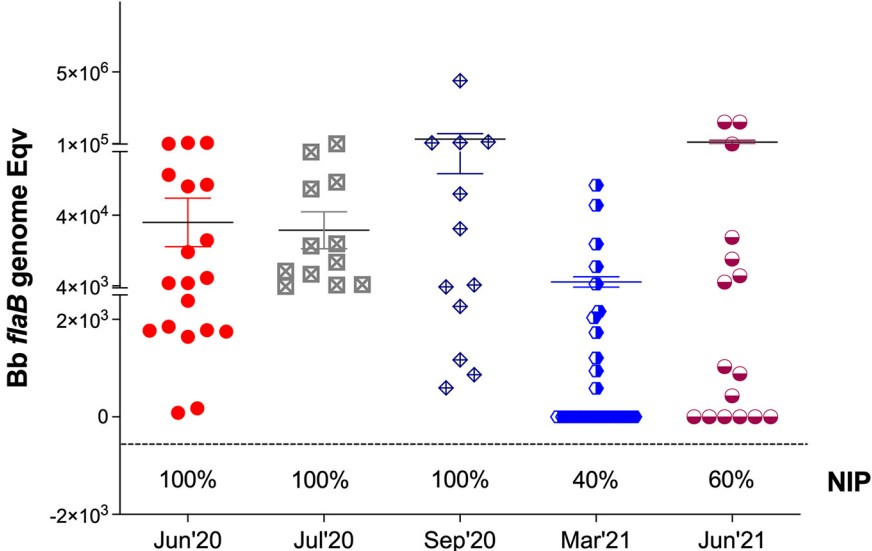

**FIG 3** *B. burgdorferi* load and nymphal infection prevalence in engorged nymphal *I. scapularis* ticks after challenge. Statistics by unpaired *t* test with Welch's correction between groups are not significant.

infection prevalence, 40–60% of ticks used in the 2021 challenges did not contain *B. burgdorferi* DNA in contrast to the ticks used for challenge in 2020 that were all infected (100%).

**Fitness of transmission of *B. burgdorferi* to naive hosts by nymphal ticks kept in questing phase for 1 year.** *I. scapularis* ticks kept in questing phase under controlled laboratory conditions were used to challenge naive mice at prime questing phase (June 2020 and July 2020) and as time progressed after prime questing, up to a year (September 2020, March 2021, and June 2021). We analyzed *B. burgdorferi* dissemination to distant tissues by qPCR of *B. burgdorferi* flaB from DNA purified from bladder and heart (Fig. 4). We found that at prime questing phase (June and July 2020) all mice (9/9, 100%) became infected after tick challenge; as time progressed post prime questing (Sep'20–Mar'21–Jun'21), 12/15 (80%) mice had evidence of *B. burgdorferi* DNA in bladder and heart tissues (Fig. 4A and B). Infection was further confirmed by the presence of IgG antibodies to *B. burgdorferi* in serum of challenged mice by immunoblot (Fig. 5). Challenges done at prime questing resulted in 9/9 (100%) infection in June and July of 2020. As time progressed away from prime questing, the efficacy of *B. burgdorferi* transmission to naive mice declined to 4 to 5/5 (80–100%) in September 2020, and to 8/10 (80%) in March and June 2021. In addition, we analyzed the viability of *B. burgdorferi* in Barbour-Stonner-Kelly (BSK) cultures of murine hearts 21 days

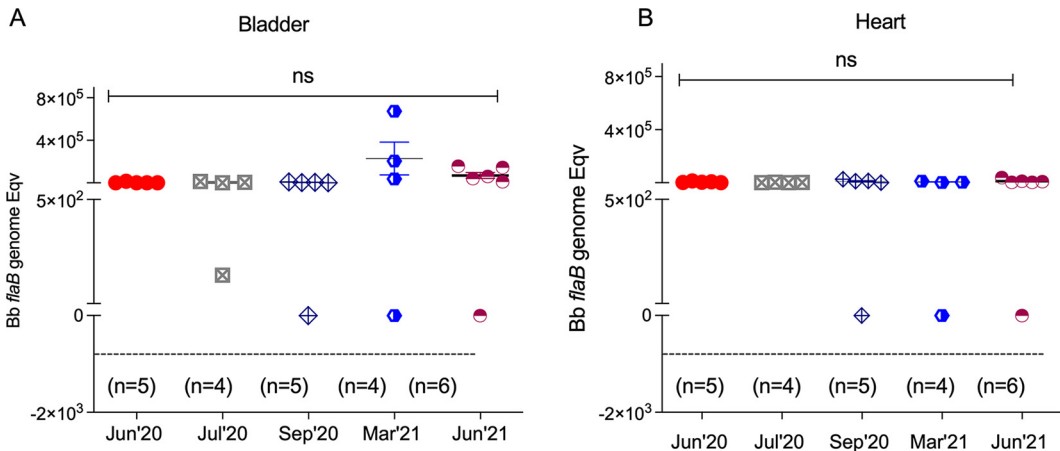

**FIG 4** Detection *of B. burgdorferi* DNA isolated from bladder and heart tissues of mice challenged with *I. scapularis* nymphs maintained in prolonged host-seeking behavior for ~1 year. Statistics by one-way ANOVA, not significant (ns).

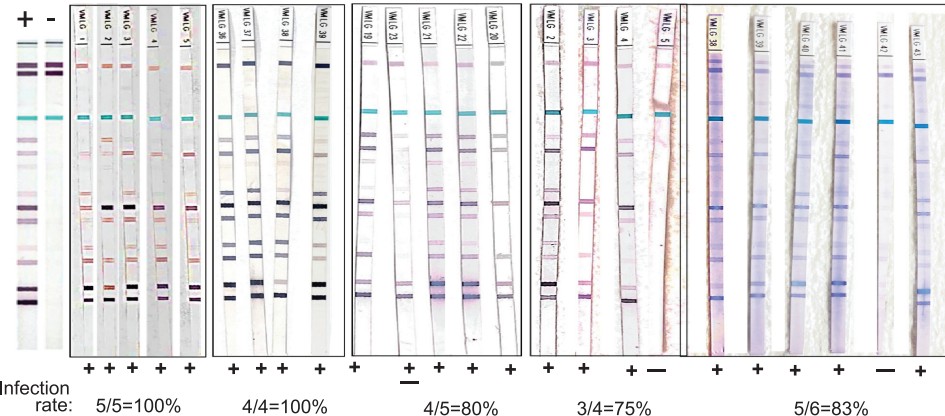

**FIG 5** Immunoblot detection of *B. burgdorferi* specific IgG in serum from mice challenged with *I. scapularis* nymphal ticks maintained in questing behavior for ~1 year. Each band represents one *B. burgdorferi* antigen. A count of 5 bands determines a positive.

post euthanasia by dark-field microscopy and qPCR. We observed growth of motile *B. burg-dorferi* and obtained positive flaB qPCR in cultures from tissues (Fig. 6) harvested from all animals that were previously positive by direct PCR from tissue DNA as well as from the mouse that had negative PCR heart and bladder in September 2020 (Fig. 4) but produced equivocal immunoblot results, mouse VMLG23 (Fig. 5). The 2 animals that had negative cultures did not have *B. burgdorferi* DNA in heart or bladder tissues (Fig. 4) and did not produce positive immunoblots (Fig. 5). After resolution by tissue culture, we confirmed that nymphal transmission of *B. burgdorferi* was 100% effective in the three challenges done in 2020 and 80% effective in the two challenges done in 2021.

## DISCUSSION

The goal of our study was to evaluate if multiple strains of *B. burgdorferi* survive in nymphal ticks in questing phase prolonged for 1 year under optimal environmental conditions and if prolonged questing affects the tick fitness to transmit *B. burgdorferi* to vertebrate hosts. We found that within the prime questing period, starting after 6 weeks of developmental diapause after molting from larva, the infection prevalence of host-seeking/questing nymphal ticks was significantly higher than dormant diapause nymphal ticks. In addition, there were no significant differences in monthly infection prevalence of ticks kept in prolonged questing phase for the duration of the experiment (1 year). More importantly, we found that although ticks kept in prolonged questing phase (12 months) by optimal environmental conditions had a slightly lower fitness (80%) for transmission of *B. burgdorferi* to naive vertebrate hosts than ticks used to challenge mice in their prime questing phase (100%) in the third quarter of the year, these differences are not likely to negatively affect the enzootic cycle of *B. burgdorferi*.

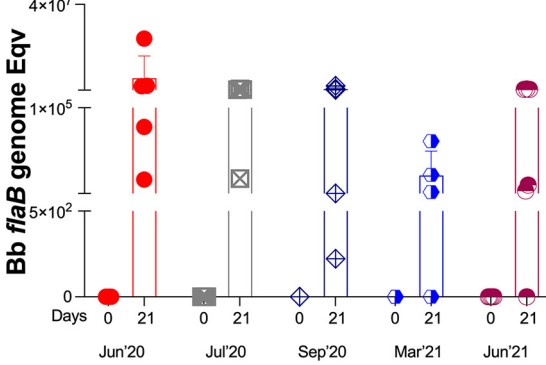

**FIG 6** Analysis of *B. burgdorferi* viability in culture from mouse tissue. Samples from heart cultures were tested on day 0 and day 21 posteuthanasia by flaB qPCR.

High mobility and the ability to grab moving objects is the expected behavior of questing nymphs that also have higher metabolic activity and lipid energy stored to be spent toward host-seeking in natural conditions (19–21). At 6 weeks post molting from infected engorged larva, when nymphal ticks usually begin host-seeking, we found a significantly higher *B. burgdorferi* nymphal infection prevalence (NIP) as well as a 10× increase in *B. burgdorferi* load in the host-seeking questing nymphs in contrast to the ticks that remained in diapause (Fig. 1). The data suggest that this excess energy and metabolic activity in questing nymphs also contributes to survival of *B. burgdorferi*. This is corroborated by findings in *Ixodes ricinus* that suggest that vector-borne pathogens can alter phenotypic traits in the vector so that micro-organism transmission is enhanced (22). They found that the level of energy reserves, hydration status, and *Borrelia* infection influence *I. ricinus*' walk over a humidity gradient (23) and that Borrelia-infected nymphal ticks have higher fat content than uninfected ticks (24).

When we checked the infection prevalence of *I. scapularis* nymphs kept in prolonged questing in the laboratory under optimal environmental conditions for 1 year (Fig. 2), we found that infection prevalence (NIP) started at 75% (6 weeks after molting, June 2020), it fluctuated randomly mostly between 60% and 80% with some outlier months at 40% or 100%, and it ended at 60% a year later (June 2021). These data suggest that under optimal environmental conditions of temperature, humidity, and natural light, host-seeking behavior can be prolonged for a least 1 year without negatively impacting the nymphal infection prevalence. Previous studies had shown that *I. scapularis* nymphs remain viable for 10 months in optimal environmental laboratory conditions of temperature, humidity, and natural light (22–24°C, 90%, 16 h/8 h light/dark) (13). Our study reproduces those results and adds information on the infection prevalence of these questing ticks and whether they can effectively transmit *B. burgdorferi* to the mammalian host. We speculate that warmer weather may facilitate survival of increased numbers of questing infectious nymphal ticks in some geographical areas where they already exist but were not previously a problem. We did measure some significant changes in *B. burgdorferi* load from month to month with an increase at the start of the experiment from June–July 2020. Since this is prime questing time (6 weeks post molting), the increase in *B. burgdorferi* load suggests an increase in tick fitness in the second month. However, in ticks subjected to prolonged questing, 7–8 months post prime questing (April/May 2021), there was a significant decline in *B. burgdorferi* load that eventually bounced back in June of 2021. In *I. ricinus*, others have shown that *B. afzelii* loads declined over 4 months in nymphal ticks produced in the laboratory under controlled laboratory conditions of 22°C and humidity >80% (25). This, however, was not observed in our studies using *I. scapularis* infected with *B. burgdorferi*. Our results looked very similar to the NIP in diapause measured 6 weeks post molting. Thus, we attribute these random reductions in NIP and *B. burgdorferi* load to behavioral diapause, which is the immediate response to hazardous environmental conditions possibly due to suboptimal humidity conditions in the glass flasks.

Another very important finding of our study was that in optimal environmental conditions of temperature, humidity, and natural daylight, *I. scapularis* nymphal ticks can maintain a prolonged questing phase for at least 1 year that competently transmits *B. burgdorferi* to naive mice. Although the fitness of infected nymphal ticks kept in prolonged questing phase was slightly reduced (from 100% to 80%, Fig. 3 to 5), it is more than enough to maintain a competent enzootic cycle and ensure effective transmission of *B. burgdorferi* to 80% of naive reservoir hosts after 1 year.

**Conclusion.** Our study shows that *B. burgdorferi* infected unfed *I. scapularis* ticks not only survived for a year in prolonged questing under optimal environmental conditions in the laboratory, they maintained a nymphal infection prevalence sufficient to effectively fulfil the enzootic cycle of *B. burgdorferi* after tick challenge. Our study raises important questions regarding prolonged survival of *B. burgdorferi* infected host-seeking nymphal *I. scapularis* that could result in increased Lyme disease incidence if conditions of temperature and especially humidity in regions currently classified as nonendemic become amenable due to global warming. Furthermore, it raises important questions regarding the mechanisms underlying survival of *B. burgdorferi* under hostile conditions.

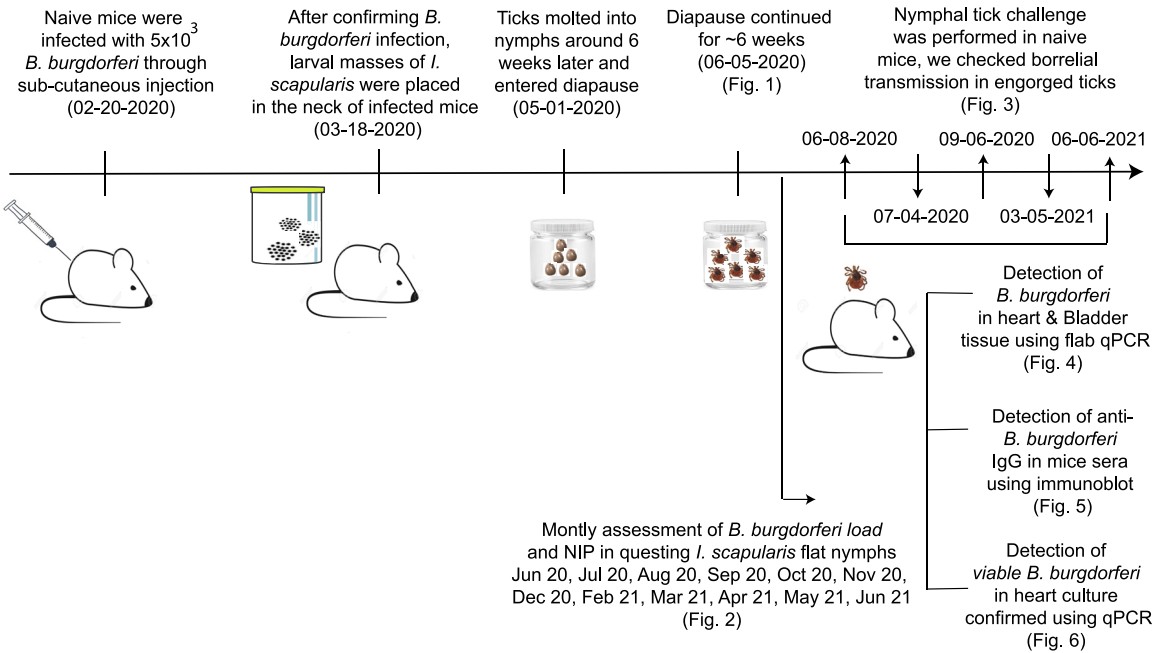

**FIG 7** Diagram of the experimental design.

## MATERIALS AND METHODS

A diagram of the experimental design is provided in Fig. 7. The study was approved by the University of Tennessee Health Science Center (UTHSC) Institutional Biosafety Committee (IBC) protocol number 12–416-R5.

**Animals.** Six-week-old female C3H/HeN mice were purchased from Charles River lab (Boston, MA). The study was conducted according to the Guide for the Care and Use of Laboratory Animals of the U.S. National Institutes of Health. The study was approved by the UTHSC Institutional Animal Care and Use Committee (IACUC) protocol number 19–0103.

**Bacterial culture and maintenance.** Multistrain cultures of *B. burgdorferi sensu stricto* were initially recovered from heart and bladder tissues of white-footed mice naturally infected with *B. burgdorferi* in 2005, 2006, 2007, and 2008. Culture samples were stored as glycerol stock at $-80°C$ for future use and were thawed, and low passage number (1–2) cultures were used for our study.

**Generation of Infected larval *I. scapularis*.** Needle challenge was performed in C3H/HeN mice with 50,000 multistrain *B. burgdorferi* cultured in our lab. Briefly, subcutaneous injection with *B. burgdorferi* was given in the neck region of C3H/HeN mice. Three weeks later, mouse serum samples were checked for *B. burgdorferi* specific antibodies. Once infection was confirmed, uninfected larvae challenge was performed. In brief, batches of clean flat larvae (Oklahoma State University) were placed on the back of the head of the mice that were restrained for 15–20 min to let the ticks attach (26). Mice were kept individually in wire bottomed cages filled with 0.5–2.5 cm water to ensure survival of fallen engorged ticks. The mice were carefully monitored 2–3 times a day to collect engorged larval ticks from the bottom of the cage for a week after challenge.

**Generation and maintenance of infected *I. scapularis* nymphs.** Engorged larval ticks (50–75) were collected and stored in clear glass vials with nylon mesh lids, and maintained at optimum temperature, relative humidity (RH), and light/dark cycles ($24 \pm 1°C$, 90% humidity, placed near a window). Ticks were closely monitored every week for about 5–6 weeks to check humidity levels and molting into flat nymphs. To ensure their survival, longevity, and fitness, waste inside the vials was removed from time to time to clean up mold and ensure that water droplets did not accumulate at the bottom of the vials. After molting, ticks were kept in the same glass vials for an additional $\sim$6 weeks until they fully exited diapause and started exhibiting host-seeking behavior (questing).

**Nymphal challenge.** Flat nymphal ticks were gently collected from the glass vials using forceps and placed on the back of the head of restrained mice (4–6 ticks per mouse). After 30 min, individual mice were placed in wire bottom cages (containing 1–2 of water) for about 1 week. The ticks were collected from the bottom of the cages after they naturally fell off and were kept frozen at $-20°C$ for PCR analysis. Three weeks after the last day of challenge, the mice were humanely euthanized and blood, heart, and bladder were collected for determination of *B. burgdorferi* dissemination by qPCR. A total of five tick challenges were performed: June 2020, July 2020, September 2020, March 2021, and June 2021.

**Assessment of nymphal infection prevalence (NIP) of *I. scapularis*.** The *B. burgdorferi* specific flaB gene was amplified using specific primers and probes from purified nymphal DNA, by qPCR. Briefly, nymphs were periodically collected from the glass container, individually crushed, and lysed with the help of Mini-Beadbeater-16 from BioSpec using sterile zirconia beads, and the DNA was purified using the DNAeasy blood and tissue kit (Qiagen, Valencia CA), according to the manufacturer's protocol. Purified DNA was stored at $-20°C$ until further use. Quantification of the total number of *B. burgdorferi* in the tick DNA was determined using a standard curve ($10^5$ serially diluted to 1) prepared with a known number of

cultured *B. burgdorferi*. *I. scapularis* specific actin primers and probes were used as a positive control. Threshold Cycle (CT) value of <38 was considered to analyze flaB amplification plot.

Analysis of transmission efficiency of *B. burgdorferi* to mice after tick challenge was done by qPCR of *B. burgdorferi* flaB from DNA purified from engorged nymphs as well as from 20–25 mg of bladder and heart.

**Determination of *B. burgdorferi* viability in BSK culture.** One half of the heart tissue was incubated in 5 mL of complete BSK (Barbour-Stonner-Kelly) media with 100× antibiotic cocktail for 3–4 weeks at 34°C. On day 0 and on day 21, BSK cultures were used to detect live spirochetes using darkfield microscopy and q-PCR.

**Detection of anti-*B. burgdorferi* antibody in serum from tick challenged mice by Western blot.** Virablot membranes precoated with *B. burgdorferi* specific antigens were purchased from Viralab (Cat#V-BBSGUS) and used for immunoblotting. Individual serum samples (1:100 dilution) were incubated with the strips for 1 h at room temperature. Each strip was washed with 1× Tris-buffered saline with Tween 20 (TBST) at least three times after every step. Secondary antibody (anti-mouse IgG) conjugated with alkaline phosphatase was added at a dilution of 1:1,000 and incubated for 30 min. Finally, strips were developed using BCIP/NBT Membrane Substrate to develop the respective bands. A pattern of 5 or more out of 10 bands (proteins with molecular weight 93, 66, 58, 45, 41, 39, 30, 28, 23, and 18 kD) was considered as positive borrelial infection.

**Statistical analysis.** Statistical analysis was done using unpaired Student's *t* test (Mann-Whitney/ Welch's correction) within two groups. Ordinary one-way ANOVA was used to compare *B. burgdorferi* load in tissue samples between tick challenged groups for different months. GraphPad Prism software was used for the statistical analysis and plotting the data set.

## ACKNOWLEDGMENTS

This work has been supported by the Public Health Service awards AI139267 (M.G.-S.) and AI155211 (M.G.-S.) from the National Institute of Allergy and Infectious Diseases (NIAID) of the National Institutes of Health (NIH) of the United States of America. The content of this manuscript is totally the responsibility of the authors and does not involve the official views of NIAID or NIH.

Conceptualization, experimental design, supervision, manuscript writing, funds acquisition: M.G.-S. Manuscript writing, data analysis: K.S. and S.K. Experimental design, investigation, data acquisition: K.S., J.F.A., and N.N.

The following authors declare potential conflicts of interest: M.G.-S. (grants from federal agencies, employment). All other authors declare no conflicts.

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
