## [Reviewer comments · Microbiology Spectrum]

Microbiology Spectrum

Infected *Ixodes scapularis* nymphs maintained in prolonged questing under optimal environmental conditions for one year can transmit *Borrelia burgdorferi* (*Borreliella* genus novum) to uninfected hosts.

Kamalika Samanta, Jose Azevedo, Nisha Nair, Suman Kundu, and Maria Gomes-Solecki

Corresponding Author(s): Maria Gomes-Solecki, University of Tennessee Health Science Center

Review Timeline:

Submission Date:	April 18, 2022
Editorial Decision:	June 9, 2022
Revision Received:	June 22, 2022
Accepted:	June 26, 2022

Editor: Catherine Brissette

Reviewer(s): Disclosure of reviewer identity is with reference to reviewer comments included in decision letter(s). The following individuals involved in review of your submission have agreed to reveal their identity: Fukai Bao (Reviewer #1)

Transaction Report:

DOI: <https://doi.org/10.1128/spectrum.01377-22>

June 9, 2022

Dr. Maria Gomes-Solecki
University of Tennessee Health Science Center
Microbiology, Immunology and Biochemistry
858 Madison Ave
Memphis, TN 38163

Re: Spectrum01377-22 (*Questing I. scapularis* ticks maintained for one year under optimal environmental conditions transmit *B. burgdorferi* to uninfected mice)

Dear Dr. Maria Gomes-Solecki:

Reviewers were mainly positive and feel this should add to the field, but please address their concerns.

Link Not Available

Sincerely,

Catherine Brissette

Journals Department
Reviewer comments:

Reviewer #1 (Comments for the Author):

In this manuscript, authors produced nymphal ticks in the laboratory under controlled temperature, humidity and natural light/dark cycle conditions to allow them to remain in host-seeking (questing) phase for 1 year; then analyzed differences in *B. burgdorferi* infection prevalence of questing and diapause nymphs at 6 weeks post-molting (prime questing) as well as differences in infection prevalence of questing nymphs maintained under prolonged environmental induced questing over 12 months (prolonged questing); Lastly, analyzed the fitness of nymphal ticks subjected to environmental induced questing in transmission of *B. burgdorferi* to naïve mice over the course of the year.
This work is some interesting, but some questions exist still.

1. Abstract has a bad format which has so many introductions, but not contained logical methods, results, and conclusion.
2. Authors should provide a biosafety statement and permission.

Reviewer #2 (Comments for the Author):

The authors hypothesize that infected *I. scapularis* nymphs that retain host-seeking behavior under optimal environmental conditions are fit to fulfill their role in transmission in the *B. burgdorferi* enzootic cycle. To test this hypothesis, they produced nymphal ticks in the laboratory under controlled temperature (22-25°C), humidity (80-90%) and natural daylight cycle conditions to allow them to retain host-seeking/questioning behavior for ~1 year. They analyzed differences in *B. burgdorferi* infection prevalence in questing and diapause nymphs at 6 weeks post-molting (prime questing) as well as differences in infection prevalence of questing nymphs maintained under prolonged environmentally induced questing over 12 months (prolonged questing). They also analyzed the fitness of nymphal ticks subjected to prolonged questing in transmission of *B. burgdorferi* to naïve mice over the course of the year. They found that infected unfed *I. scapularis* nymphal ticks maintained under optimal environmental conditions in the laboratory survived for a year in a developmental state of prolonged questing (host-seeking), they retained an infection prevalence sufficient to effectively fulfill transmission of *B. burgdorferi* to uninfected mice after tick challenge. The authors believe their study is important for understanding and possibly modeling Lyme disease expansion into former non-endemic regions due to global warming. (From the Abstract).

This is an interesting, focused and hypothesis-based work demonstrating that infected *I. scapularis* nymphs retain extended (over a year) host-seeking behavior under optimal environmental conditions and are fit to fulfill their transmission role in the enzootic cycle of *B. burgdorferi*. The demonstration of the ability of *B. burgdorferi* to survive long periods of time (over a year) in unfed ticks and conserve its infectiousness is also a very relevant finding to understanding the epidemiology of Lyme disease. It also triggers questions regarding the mechanism(s) underlying survival of the bacteria in apparently hostile conditions. The relevance of these important experimental results is that they may be useful to understanding the evolving epidemiology of Lyme disease and the diffusion of the disease from endemic to non-endemic areas under a background of climate change.

The use by the authors of infectious *B. burgdorferi* *sensu stricto* mixed cultures recovered from wild footed mice organs is an important plus of these experiments as they recapitulate more closely the natural situation where the pathogen infects animals with a mixed population. The experiments to infect mice and *I. scapularis* tick larva were carried out using standard procedures; adequate controls for infection were included. The authors took special care in the generation of prime and prolonged questing nymphs and confirmation and quantification of their infection with *B. burgdorferi* by qPCR of the *flaB* gene and controls throughout the duration of the experiments. . Generation and assessment of infection of these nymphs with *B. burgdorferi* is a crucial step in these studies and the care with which the authors carried them out gives confidence on their results and their analysis. The molecular and immunological methods used to measure the efficiency of transmission and infection of the mice were adequate and provide further trust on the results obtained. Figure 7 clearly represents the protocol of the experiments, while the other the figures provide an excellent illustration of the results obtained.

The authors clearly demonstrate that host-seeking ticks had *B. burgdorferi* at higher frequency and concentration than non-host-seeking ticks and that the difference in frequency was significant and remained higher for over a year. More importantly, the nymphs so infected were proficient in their ability to transmit *B. burgdorferi* to naïve mice as determined by qPCR, tissue culture of infected mice and antibody response. Thus, as the authors state, these results "raise important questions regarding prolonged survival of *B. burgdorferi*-infected host-seeking nymphal *I. scapularis* that could result in increased Lyme disease incidence if conditions of temperature and especially humidity in regions currently classified as non-endemic become amenable due to global warming."

This being said, I believe the authors should also articulate and describe a scenario of how these questing tick nymphs highly infected with *B. burgdorferi* will help to disseminate Lyme disease to terrains virgins of the disease. Will they migrate carried by rodent or birds? Or will the warmer weather facilitate their presence and numbers in some geographical areas where they already exist but were not previously a problem? In other words, how do the authors envision these ticks spreading Lyme disease to untouched areas. The authors should also expand their discussion as to how they see the tick metabolism interacting with *B. burgdorferi* metabolism to ensure persistence of viable *B. burgdorferi* for over a year and be efficiently transmitted to mice hosts (lines 191 and 192). In this context, what is their explanation for the NIP differences between diapause and questing ticks?

In the title *Ixodes* and *Borrelia* should be spelled fully, and a description of the statistical methods used to analyze the data should be added to the Material and Methods section; their description in the figure legends is not sufficient. What is the meaning of the abbreviation NS in Figure 2? Abbreviations used less than 3 times should not be employed and should be written out.

Staff Comments:

Preparing Revision Guidelines

Please return the manuscript within 60 days; if you cannot complete the modification within this time period, please contact me. If you do not wish to modify the manuscript and prefer to submit it to another journal, please notify me of your decision immediately so that the manuscript may be formally withdrawn from consideration by Microbiology Spectrum.

Response to Reviewers

Re: **Spectrum**01377-22 (*Questing I. scapularis* ticks maintained for one year under optimal environmental conditions transmit *B. burgdorferi* to uninfected mice)

Reviewer comments:

Reviewer #1

This work is some interesting, but some questions exist still.

1. Abstract has a bad format which has so many introductions, but not contained logical methods, results, and conclusion.

The abstract was reformatted to address these concerns.

2. Authors should provide a biosafety statement and permission.

This was added to the first sentence on the Methods section.

Reviewer #2 (Comments for the Author):

This is an interesting, focused and hypothesis-based work demonstrating that infected *I. scapularis* nymphs retain extended (over a year) host-seeking behavior under optimal environmental conditions and are fit to fulfill their transmission role in the enzootic cycle of *B. burgdorferi*. The demonstration of the ability of *B. burgdorferi* to survive long periods of time (over a year) in unfed ticks and conserve its infectiousness is also a very relevant finding to understanding the epidemiology of Lyme disease. It also triggers questions regarding the mechanism(s) underlying survival of the bacteria in apparently hostile conditions. The relevance of these important experimental results is that they may be useful to understanding the evolving epidemiology of Lyme disease and the diffusion of the disease from endemic to non-endemic areas under a background of climate change.

I added the underlined concept to the discussion.

The use by the authors of infectious *B. burgdorferi* *senso stricto* mixed cultures recovered from wild footed mice organs is an important plus of these experiments as they recapitulate more closely the natural situation where the pathogen infects animals with a mixed population. The experiments to infect mice and *I. scapularis* tick larva were carried out using standard procedures; adequate controls for infection were included. The authors took special care in the generation of prime and prolonged questing nymphs and confirmation and quantification of their infection with *B. burgdorferi* by qPCR of the *flaB* gene and controls throughout the duration of the experiments. . Generation and assessment of infection of these nymphs with *B. burgdorferi* is a crucial step in these studies and the care with which the authors carried them out gives confidence on their results and their analysis. The molecular and immunological methods used to measure the efficiency of transmission and infection of the mice were adequate and provide further trust on the results obtained. Figure 7 clearly represents the protocol of the experiments, while the other the figures provide an excellent illustration of the results obtained.

I highlighted the fact that we used multiple strains of *B. burgdorferi* in the study in the

first sentence of the discussion.

The authors clearly demonstrate that host-seeking ticks had *B. burgdorferi* at higher frequency and concentration than non-host-seeking ticks and that the difference in frequency was significant and remained higher for over a year. More importantly, the nymphs so infected were proficient in their ability to transmit *B. burgdorferi* to naïve mice as determined by qPCR, tissue culture of infected mice and antibody response. Thus, as the authors state, these results "raise important questions regarding prolonged survival of *B. burgdorferi*-infected host-seeking nymphal *I. scapularis* that could result in increased Lyme disease incidence if conditions of temperature and especially humidity in regions currently classified as non-endemic become amenable due to global warming."

This being said, I believe the authors should also articulate and describe a scenario of how these questing tick nymphs highly infected with *B. burgdorferi* will help to disseminate Lyme disease to terrains virgins of the disease. Will they migrate carried by rodent or birds? Or will the warmer weather facilitate their presence and numbers in some geographical areas where they already exist but were not previously a problem? In other words, how do the authors envision these ticks spreading Lyme disease to untouched areas. The authors should also expand their discussion as to how they see the tick metabolism interacting with *B. burgdorferi* metabolism to ensure persistence of viable *B. burgdorferi* for over a year and be efficiently transmitted to mice hosts (lines 191 and 192). In this context, what is their explanation for the NIP differences between diapause and questing ticks?

I added a little more in the discussion lines 192-196 and lines 209-211 to include these excellent comments:

192-196: This is corroborated by findings in *Ixodes ricinus* that suggest that vector-borne pathogens can alter phenotypic traits in the vector so that microorganism transmission is enhanced (Herrmann, 2015). They found that the level of energy reserves, hydration status and *Borrelia* infection influence *I. ricinus*' walk over a humidity gradient (Herrmann, 2012) and that *Borrelia*-infected nymphal ticks have higher fat content than uninfected ticks (Herrmann, 2013).

209-211: We speculate that warmer weather may facilitate survival of increased numbers of questing infectious nymphal ticks in some geographical areas where they already exist but were not previously a problem.

In the title *Ixodes* and *Borrelia* should be spelled fully, and a description of the statistical methods used to analyze the data should be added to the Material and Methods section;

Ixodes and *Borrelia* were included in title and a description of statistical methods were added in the methods section.

their description in the figure legends is not sufficient. What is the meaning of the abbreviation NS in Figure 2? Abbreviations used less than 3 times should not be employed and should be written out.

These issues were addressed in the revised manuscript.

June 26, 2022

Dr. Maria Gomes-Solecki
University of Tennessee Health Science Center
Microbiology, Immunology and Biochemistry
858 Madison Ave
Memphis, TN 38163

Re: Spectrum01377-22R1 (Infected Ixodes scapularis nymphs maintained in prolonged questing under optimal environmental conditions for one year can transmit Borrelia burgdorferi (Borreliella genus novum) to uninfected hosts.)

Dear Dr. Maria Gomes-Solecki:

This work raises important questions regarding tick borne pathogens and climate change. The very minor concerns of the reviewers were addressed.

Your manuscript has been accepted, and I am forwarding it to the ASM Journals Department for publication. You will be notified when your proofs are ready to be viewed.

Sincerely,

Catherine Brissette
Editor, Microbiology Spectrum
